# Crosstalk between Host Genome and Metabolome among People with HIV in South Africa

**DOI:** 10.3390/metabo12070624

**Published:** 2022-07-06

**Authors:** Chang Liu, Zicheng Wang, Qin Hui, Yiyun Chiang, Junyu Chen, Jaysingh Brijkumar, Johnathan A. Edwards, Claudia E. Ordonez, Mathew R. Dudgeon, Henry Sunpath, Selvan Pillay, Pravi Moodley, Daniel R. Kuritzkes, Mohamed Y. S. Moosa, Dean P. Jones, Vincent C. Marconi, Yan V. Sun

**Affiliations:** 1Department of Epidemiology, Rollins School of Public Health, Emory University, Atlanta, GA 30322, USA; chang.liu2@emory.edu (C.L.); qhui@emory.edu (Q.H.); ellen.chiang@emory.edu (Y.C.); junyu.chen@emory.edu (J.C.); 2College of Arts and Sciences, Emory University, Atlanta, GA 30322, USA; wendy.wang@emory.edu; 3Nelson R Mandela School of Medicine, University of KwaZulu-Natal, Durban 4041, South Africa; jbrijkumar@gmail.com (J.B.); henrysunpath@yebo.co.za (H.S.); selvan@adrenergy.com (S.P.); moosay@ukzn.ac.za (M.Y.S.M.); 4Department of Biostatistics and Bioinformatics, Rollins School of Public Health, Emory University, Atlanta, GA 30322, USA; alex.edwards@emory.edu; 5School of Medicine, Emory University, Atlanta, GA 30322, USA; mdudgeo@emory.edu (M.R.D.); dpjones@emory.edu (D.P.J.); vcmarco@emory.edu (V.C.M.); 6Lincoln International Institute for Rural Health, School of Health and Social Care, University of Lincoln, Lincoln LN6 7TS, UK; 7Hubert Department of Global Health, Rollins School of Public Health, Emory University, Atlanta, GA 30322, USA; claudia.ordonez@emory.edu; 8National Health Laboratory Service, School of Laboratory Medicine and Medical Sciences, University of KwaZulu-Natal, Durban 4011, South Africa; moodleyp36@ukzn.ac.za; 9Brigham and Women’s Hospital, Harvard Medical School, Boston, MA 02115, USA; dkuritzkes@bwh.harvard.edu; 10Emory Vaccine Center, Atlanta, GA 30322, USA

**Keywords:** GWAS, metabolome, HIV, African

## Abstract

Genome-wide association studies (GWAS) of circulating metabolites have revealed the role of genetic regulation on the human metabolome. Most previous investigations focused on European ancestry, and few studies have been conducted among populations of African descent living in Africa, where the infectious disease burden is high (e.g., human immunodeficiency virus (HIV)). It is important to understand the genetic associations of the metabolome in diverse at-risk populations including people with HIV (PWH) living in Africa. After a thorough literature review, the reported significant gene–metabolite associations were tested among 490 PWH in South Africa. Linear regression was used to test associations between the candidate metabolites and genetic variants. GWAS of 154 plasma metabolites were performed to identify novel genetic associations. Among the 29 gene–metabolite associations identified in the literature, we replicated 10 in South Africans with HIV. The *UGT1A* cluster was associated with plasma levels of biliverdin and bilirubin; *SLC16A9* and *CPS1* were associated with carnitine and creatine, respectively. We also identified 22 genetic associations with metabolites using a genome-wide significance threshold (*p*-value < 5 × 10^−8^). In a GWAS of plasma metabolites in South African PWH, we replicated reported genetic associations across ancestries, and identified novel genetic associations using a metabolomics approach.

## 1. Introduction

Genome-wide associations studies (GWAS), which involve investigations on associations between genetic architecture and complex disease traits, have been widely conducted to understand how the genetic profile contributes to various diseases [1]. High-resolution metabolomics can measure large-scale profiles of small molecules in biological samples, which enables metabolome-wide association studies (MWAS) to understand a variety of diseases on a molecular basis [2]. The traditional GWAS has been hindered by the phenotyping of exposures and risk factors that are unmeasured or even unknown. By combining high-throughput phenotyping metabolomics data with GWAS (mGWAS), we have broadened our insight into many complex diseases and traits [3].

Early mGWAS research found that common single nucleotide polymorphisms (SNP) can account for as much as 12% of the variance in homeostasis of the human metabolic profile [4]. In recent years, a number of studies of blood and urine metabolites have been conducted, revealing many robust and reproducible associations between genetic variants and metabolites, with many involving enzyme catalysis and transporter proteins [5,6]. With a goal of building an integrated atlas of genomics and metabolomics, many investigations have focused on general populations of European descent [7,8], and only a few have focused on African Americans [9,10]. For example, the *UGT1A1* gene encodes the enzyme in the bilirubin metabolism, and its association with bilirubin and biliverdin has been successfully established [9,11]. This particular locus is also involved in the metabolic pathways linking to several human immunodeficiency virus (HIV) medications; however, there have been no studies focusing on individuals of African ancestry who have HIV infection living in South Africa, which would permit an exploration of environmental and genetic influences on metabolic processes. In this study of people with HIV (PWH) living in South Africa who were antiretroviral therapy (ART) naive, we aimed to assess the generalizability of previously reported findings and explore the novel associations between genetic variants and metabolites in this unique cohort of genetic ancestry, geography, and health condition.

## 2. Results

### 2.1. Characteristics of Study Participants

All 490 participants were South Africans, with 305 from RK Khan Hospital and 185 from Bethesda Hospital. The mean age was 34.4 (standard deviation [SD] 10.0) years, and 312 (63.7%) were female (Table 1). The ancestry map of the South African study participants compared to the 1000 Genome Project [12] reference panel showed a well-defined South African genetic profile partially intersecting with African (AFR) ancestry but distinct from other ancestries (Figure 1).

### 2.2. Candidate Gene–Metabolite Associations

From the 24 reported GWAS of blood-based metabolic profiles (Figure 2, Table 2), we searched for the metabolites analyzed and annotated in the present study. We identified a total of 29 candidate SNP-metabolite associations, including 18 independent genetic loci and 14 metabolites from previous studies, and tested these associations in the South African cohort (Table 3). Five metabolites including citrulline, glutamine, histidine, serine, and urate, were identified from two different liquid chromatography platforms and were both analyzed, resulting in 36 replications of SNP-metabolite associations. Among the 36, a total of 24 were consistent in direction regardless of statistical significance, 7 were inconsistent, and 5 could not be compared due to lack of beta coefficients in the publications. A total of seven were both consistent in direction and statistically significant. The *UGT1A1/UGT1A* cluster locus showed robust association, with the T allele of the lead SNP rs887829 associated with higher bilirubin (0.31 ± 0.07, *p* = 3.38 × 10^−6^) and biliverdin (0.39 ± 0.07, *p* = 1.04 × 10^−8^) levels. The G allele of rs1171617 in *SLC16A9* and A allele of rs7422339 in *CPS1* were associated with lower carnitine (−0.23 ± 0.07, *p* = 0.0014) and higher creatine (0.19 ± 0.07, *p* = 0.0045), respectively (Figure 3, Table 3).

### 2.3. GWAS

The 154 inflation factors (lambda) of GWAS had a range of 0.98–1.02 (Appendix A). The Q–Q plots and Manhattan plots are presented in Appendix A. A total of 22 genetic associations with metabolites passed the genome-wide significance level of *p* < 5 × 10^−8^ (Table 4). The rs887829 variant of *UGT1A1/UGT1A* cluster locus was one of the top associations in the GWAS of biliverdin. Several novel loci were identified, such as the G allele of rs1401798 associated with higher hypoxanthine (0.37 ± 0.06, *p* = 8.80 × 10^−10^), and the G allele of rs6874865 associated with lower 3-methyl-2-oxindole (−0.48 ± 0.08, *p* = 8.66 × 10^−9^).

## 3. Discussion

Low- and middle-income countries, such as South Africa, are facing unique public health challenges with high prevalence of infectious disease, and the growing burden of noncommunicable diseases [29]. However, most genomic and metabolic studies have been conducted on individuals of European ancestry from high-income countries. Due to different genetic ancestries and environmental exposures, the genomic and metabolomic profiles and associations with European ancestry may not apply to people with non-European ancestry from low- and middle-income countries. The present mGWAS in South Africans provides one of the first catalogs to start filling the knowledge gap in diverse global populations. Understanding the genomic and metabolic characteristics from underrepresented populations is critically needed to identify molecular mechanisms driving risks for chronic infections and noncommunicable diseases. The -omics findings will also shed light on novel precision medicine applications for at-risk populations from low- and middle-income countries to reduce health disparities.

In this mGWAS conducted among 490 PWH from Africa, we replicated previously reported genetic associations primarily from European individuals living in high-income countries. The *UGT1A1/UGT1A* cluster locus was associated with biliverdin and bilirubin, which are both bile pigments formed during the breakdown of hemoglobin in red blood cells. Biliverdin can be converted as unconjugated bilirubin by biliverdin reductase before being further metabolized. The *UGT1A1* gene encodes the bilirubin uridine diphosphate-glucuronosyltransferase, which is the enzyme responsible for conjugation, detoxification, and clearance of bilirubin, and it has been found among European, African, and Asian populations; mutations of the gene may cause unconjugated hyperbilirubinemia [30,31,32,33,34]. Further, elevated bilirubin levels were found associated with lower risk in cardiovascular events among PWH [35], and the *UGT1A1* gene is particularly of interest in future studies of HIV, since it was involved in the metabolic pathways linking to several HIV medications, such as Bictegravir, Cabotegravir, Dolutegravir, Elvitegravir, and Raltegravir (https://clinicalinfo.hiv.gov/en/guidelines/hiv-clinical-guidelines-adult-and-adolescent-arv/characteristics-integrase-inhibitors, accessed on 15 June 2022). The *CPS1* locus encodes a mitochondrial enzyme responsible for carbamoyl phosphate synthesis from ammonia and bicarbonate with the use of adenosine triphosphate. This is the first step of the urea cycle, which can generate arginine, a critical precursor of creatine synthesis [17]. Creatine can be both naturally synthesized or ingested in a usual diet to supply energy to the muscles, and then is converted to creatinine, filtered from the blood by the glomerulus [36]. The association between *CPS1* and creatine was previously reported among participants of European descent in the Framingham Heart Study [17], and another study of European participants, which demonstrated that the locus was also associated with creatinine [37]. As another critical compound in energy metabolism, carnitine transports long-chain fatty acids into the mitochondria to be oxidized and produce energy [38,39]. The product of *SLC16A9* is a carnitine efflux transporter [8]; its association with carnitine was previously established among several studies in European populations but has not been generalized to other ancestries [8,17,22]. In addition, we found a borderline statistically significant association of serine and the *PHGDH* gene, which encodes the enzyme involved in the process of serine synthesis [7,8].

To the best of our knowledge, this is the first population study of genome-metabolome cross-talk among individuals of African ancestry living in Africa, which represents a population facing a unique public health burden of both non-communicable diseases and chronic infectious diseases. Although the relatively moderate sample size limited us from identifying more genome-wide significant association with metabolites, our results demonstrated the value of such a mGWAS in this under-represented population. Beyond several gene–metabolite associations that are generalizable across ancestries, several novel discoveries provide candidates for replication and may benefit future research endeavors in African countries. The identified gene–metabolite associations can be used as genetic instrumental variables in the Mendelian Randomization framework to investigative potential causal relationship between metabolites and disease outcomes. There are a few limitations associated with the present study. For example, the plasma samples were not collected during fasting, and information on diet was not surveyed, which may influence the abundance of certainly diet-related metabolites. A causal pathway may not be directly inferred from these observed associations. Future studies are warranted to provide guidance on translation and clinical implementations of identified genes and metabolites among the PWH population. In addition, future studies with improved metabolomic coverage and annotation will help establish a more complete atlas of genome-metabolome relationship in this unique population. With more and larger mGWAS in African populations, improved statistical power will facilitate robust findings of gene–metabolite associations to better understand molecular mechanisms, and to investigate the functional role of less frequent genetic variants on disease risk.

## 4. Materials and Methods

### 4.1. Study Participants

Ethics approval from the Biomedical Research Ethics Committee of the University of KwaZulu-Natal, the Emory University, and Mass General Brigham institutional review boards was obtained prior to the start of the study. After signed informed consent, people with HIV (PWH) who were at least 18 years of age and qualified for anti-retroviral therapy, were enrolled into the HIV AIDS Drug Resistance Surveillance Study (ADReSS). This study was based on a sub-cohort from the ADReSS participants recruited from KwaZulu-Natal, South Africa. The recruitment was conducted in 2014–2016 at two participating clinical sites, the RK Khan Hospital and Bethesda Hospital in KwaZulu-Natal, South Africa [40]. A total of 490 participants from the study with both genomics and metabolomics data available were included in the analysis. Full summary statistics of the results presented in the study are available upon request. Individual level dataset underlying this study can be requested and shared in compliance with the informed consent and IRB approval from participating institutes.

### 4.2. Genotyping and Imputation

Genotyping was performed on DNA extracted from blood samples of 998 participants, using the Illumina Global Screening Array. Genotype imputation for the 998 samples was performed based on the TOPMed reference panel in the genome build GRCh38 [41], using 590,511 genotyped SNPs. A total of 8,568,848 genetic variants with imputation quality R^2^ > 0.5, genotype call rate >95%, Hardy–Weinberg equilibrium *p* > 10^−6^, and minor allele frequency >0.05 entered analysis. Total of 490 participants with African ancestry also had metabolomics data from plasma samples after quality control procedures.

### 4.3. Metabolomic Profiling

The high-resolution metabolic profiling was performed using liquid chromatography with high-resolution mass spectrometry (Thermo Scientific Fusion, Waltham, MA, US) based on an established protocol [42,43,44]. Plasma samples collected from participants prior to antiretroviral therapy initiation were stored at −80 Celsius before thawing for analysis by liquid-chromatography mass spectrometry (Thermo Scientific Fusion, Waltham, MA, USA). Thawed plasma was treated with acetonitrile containing an internal isotopic standard mix [44,45], then centrifuged for 10 min at 4 Celsius to remove protein. The remaining supernatant was then maintained in a Dionex Ultimate 3000 autosampler at 4 Celsius until analysis. Two liquid chromatography strategies were used, including Supelco Ascentis Express HILIC 100 × 2.1 mm (53939-U) columns and Higgins C18100 × 2.1 mm (TS-1021-C185) columns with positive and negative ionization mode, respectively. All samples were analyzed in triplicate, with the standard reference samples of National Institute of Standards and Technology 1950 analyzed at the beginning and end of the analysis, and pooled plasma reference samples inserted at the beginning and end of each batch of 20 study samples [46]. Metabolic feature extraction was performed using apLCMS and xMSanalyzer [47,48]. Batch effect was corrected using Combat [49]. A metabolic feature was defined by the mass-to-charge ratio (m/z) ranging from 85 to 1275, and retention time up to 5 min. Pearson correlation coefficient ≥0.7 within triplicates of the metabolic feature intensities were retained, and the triplicates of intensities were median summarized based on the non-zero readings. Then, the zero readings of each metabolic feature were imputed as random numbers below the minimum intensity across all study samples. Annotations of metabolic features were matched to an in-house library of confirmed metabolites, which were of Level 1 identification confidence per established criteria [50,51]. The matching of annotations allowed m/z differences of 10 parts-per-million and retention time differences of 30 s. After removing metabolic features with multiple matches of annotations, a total of 154 metabolites were successfully and uniquely matched, as shown in Appendix A.

### 4.4. Literature Review

The diagram of the literature review is shown in Figure 2. We searched PubMed for the key words “GWAS” and “Metabolome” and found 566 articles from 2008–2022. Among those, a total of 386 articles were human studies in the English language. We reviewed these articles and found 24 blood-based GWAS of metabolic profile in Table 2. We then identified statistically significant gene–metabolite associations to pursue replication in this present study of South Africans.

### 4.5. Statistical Analysis

Genetic background of the 490 study participants were analyzed coupled with the 1000 Genome Project [12] reference panel, which included African (AFR), American (AMR), East Asian (EAS), European (EUR), and South Asian (SAS). Principal components (PCs) were computed, and the ancestry map was generated based on the top two PCs that explained the most variance. For the analysis of gene–metabolite associations, the intensities of metabolites were rank-based inverse normal transformed to achieve normality. Linear regression and additive genetic models were used to test the associations between the candidate metabolites and SNPs, controlling for age, gender, study sites, and the top ten PCs, which were derived from PC analysis of the study participants. We also performed GWAS for each of the 154 metabolites identified to explore novel genetic loci, using the same statistical model. For candidate gene–metabolite association analysis, we considered *p* < 0.05 as statistically significant, and the genome-wide significance threshold *p* < 5 × 10^−8^ was used for GWAS. All analyses were conducted in R version 4.0.2 (Vienna, Austria. URL Available online: https://www.R-project.org/ (accessed on 15 June 2022) and the PLINK software [52].

## 5. Conclusions

Among a cohort of PWH living in South Africa prior to ART initiation, we identified genetic associations with metabolites across ancestries as well as unique to ancestry group, which supports the importance of genomic association studies in diverse ancestries. Findings in gene–metabolite associations from diverse ancestries and geographic regions can provide insights to the disease risk at the molecular level and reduce health disparities for underrepresented populations.

## Figures and Tables

**Figure 1 metabolites-12-00624-f001:**
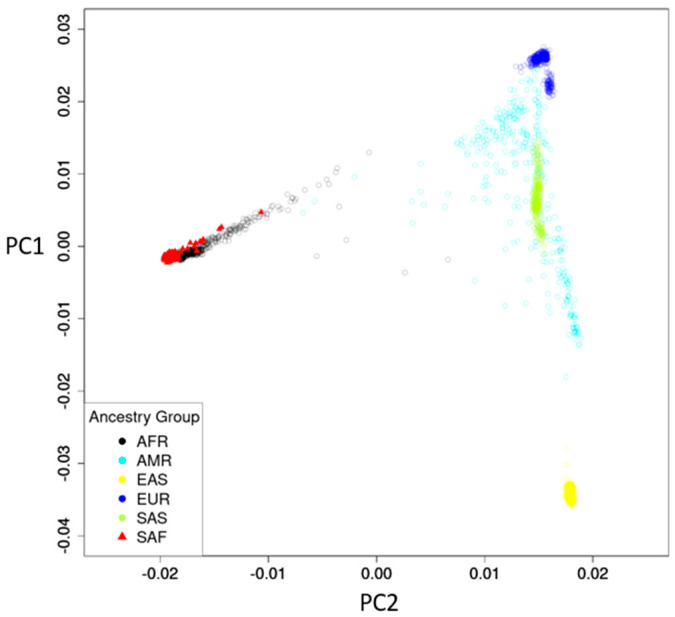
Ancestry map of study participants and reference panel. Reference panel of 1000 Genome project: AFR, African; AMR, American; EAS, East Asian; EUR, European; SAS, South Asian. Study participants: SAF, South African.

**Figure 2 metabolites-12-00624-f002:**
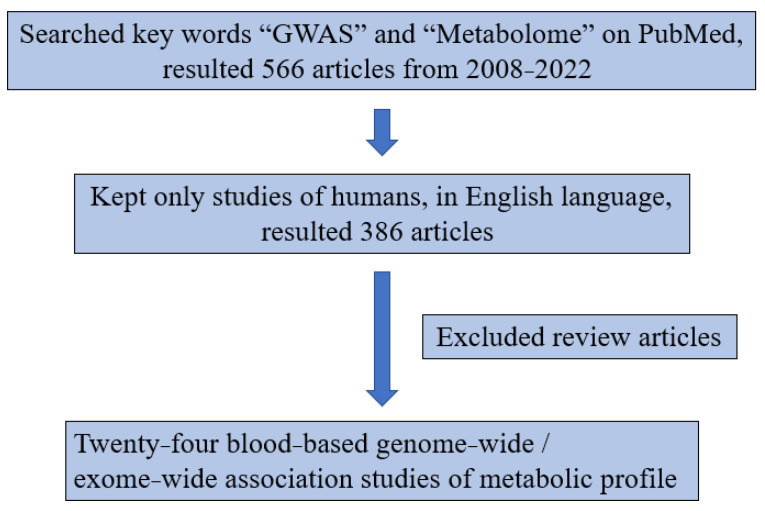
Diagram of literature review.

**Figure 3 metabolites-12-00624-f003:**
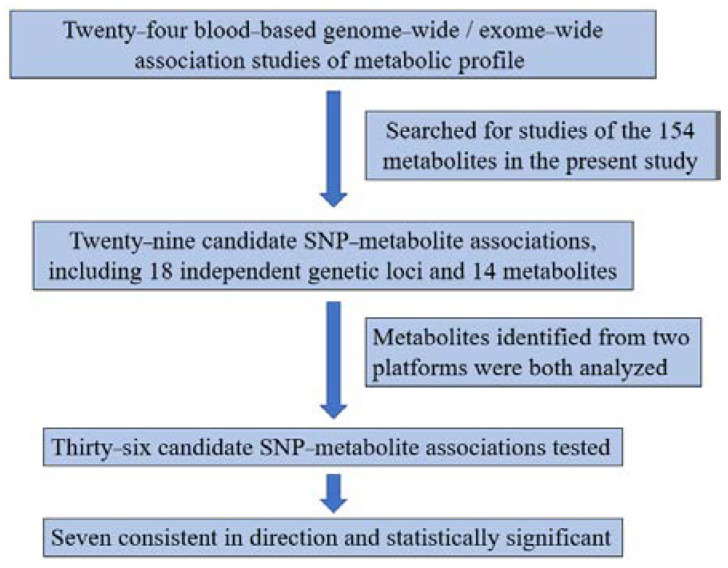
Diagram of the candidate gene–metabolite associations.

**Table 1 metabolites-12-00624-t001:** Baseline characteristics of the study cohort.

Characteristic		Overall *n* = 490	RK Khan Hospital *n* = 305	Bethesda Hospital *n* = 185
Ethnicity	Zulu (%)	370 (75.5)	189 (62.0)	181 (97.8)
	Xhosa (%)	78 (15.9)	78 (25.6)	0 (0.0)
	Other (%)	42 (8.6)	38 (12.5)	4 (2.2)
Female (%)		312 (63.7)	192 (63.0)	120 (64.9)
Age in years (SD)		34.4 (10.0)	34.1 (9.5)	34.7 (10.8)
Education in years (SD)		9.3 (3.4)	9.7 (2.9)	8.8 (4.0)
CD4 count/μL (SD)		405.034 (237.176)	427.027 (242.791)	366.286 (223.576)

**Table 2 metabolites-12-00624-t002:** Articles selected for the candidate gene–metabolite associations.

Year	First Author	Sample	Number of Metabolites	Sample Size, Country/Region	Genetic Ancestry
2008	Christian Gieger [4]	serum	363	284, Germany	European
2010	Thomas Illig [13]	serum	163	Discovery: 1809, Germany Validation: 422, UK	European
2011	Karsten Suhre [8]	serum	276	Cohort 1: 1768, Germany Cohort 2: 1052, UK	European
2012	Johannes Kettunen [14]	serum	117	8330, Finland	European
2012	Michael Inouye [15]	serum	130	Cohort 1: 1905, Finland Cohort 2: 4703, Finland	European
2012	Jan Krumsiek [16]	serum	517	1768, Germany	European
2013	Eugene P Rhee [17]	plasma	217	2076, US	European
2014	So-Youn Shin [7]	plasma and serum	486	Cohort 1: 6056, UK Cohort 2: 1768, Germany	European
2014	Bing Yu [9]	serum	308	1260, US	African (African Americans)
2014	Janina S Ried [18]	serum	344	Discovery: 1809, Germany Validation: 843, UK	European
2015	Ayşe Demirkan [19]	serum	42	2118, Netherlands	European
2015	Harmen H M Draisma [5]	serum	129	Discovery: 7478, Netherlands, Germany, Australia, Estonia, UK Validation: 1182, Germany	European
2016	Eugene P Rhee [20]	plasma	217	Discovery: 2076, US Validation: 1528, US	European
2016	Johannes Kettunen [21]	plasma and serum	123	24925, Europe	European
2016	Idil Yet [22]	serum	648	1001, UK	European
2017	Tao Long [23]	serum	644	1960, UK	European
2018	Yong Li [6]	serum and urine	serum: 139 urine: 41	1168, Germany	European
2018	Noha A. Yousri [24]	plasma	826	Discovery: 614, Qatar Validation: 382, Qatar	Middle Eastern
2018	Tanya M Teslovich [25]	serum	9	Discovery: 8545, Finland Validation: 2591, Finland	European
2019	Rubina Tabassum [26]	plasma	141	2181, Finland, UK	European
2020	Elena V Feofanova [27]	serum	640	Discovery: 3926, US Validation: 1509, US; 1960, UK	Discovery: Hispanic Validation: European
2021	Shengyuan Luo [10]	serum	652	Discovery: 619, US Validation: 818, US	African (African Americans)
2021	Eric L Harshfield [28]	serum	Cohort 1: 340 Cohort 2: 399	Cohort 1: 5662, Pakistan Cohort 2: 13,814, UK	Cohort 1: South Asian Cohort 2: European
2022	Eugene P Rhee [11]	plasma	537	822 White, 687 Black, US	European, African (African Americans)

**Table 3 metabolites-12-00624-t003:** Previously reported genetic associations with metabolites.

Metabolite	Previous Literature									ADReSS			
	First Author	rsID	Chr.	Pos. (GRCh38)	Nearest Gene	Effect/ Non-Effect Allele	Effect Allele Freq. *	Beta (SE) **	*p*	Effect Allele Freq.	Beta	SE	*p*
Bilirubin	Eugene P Rhee [11]	rs7567229	2	233703893	*UGT1A6-10*	A/C	0.31	0.31 (0.04)	2.7 × 10^−15^	0.60	0.02	0.06	0.7778
	Eugene P Rhee [11]	rs887829	2	233759924	*UGT1A3-10*	T/C	0.46	0.38 (0.05)	1.4 × 10^−13^	0.40	0.31	0.07	**3.38 × 10^−6^**
	Bing Yu [9]						0.44	NA	1 × 10^−17^				
	Shengyuan Luo [10]	rs4148325	2	233764663	*UGT1A1*, *UGT1A3-10*	T/C	0.45	0.36	3.82 × 10^−12^	0.40	0.30	0.07	**5.72 × 10^−6^**
Biliverdin	Shengyuan Luo [10]	rs1976391	2	233757337	*UGT1A3-10*	G/A	0.45	0.42	3.69 × 10^−17^	0.40	0.39	0.07	**1.04 × 10^−8^**
	So-Youn Shin [7]	rs887829	2	233759924	*UGT1A3-10*	T/C	0.34	0.113 (0.004)	2.50 × 10^−168^	0.40	0.39	0.07	**1.04 × 10^−8^**
	Bing Yu [9]						0.44	NA	8× 10^−23^				
	Eugene P Rhee [11]	rs4148325	2	233764663	*UGT1A1*, *UGT1A3-10*	T/C	0.33	0.27 (0.03)	5.7 × 10^−19^	0.40	0.38	0.07	**2.36 × 10^−8^**
Carnitine	So-Youn Shin [7]	rs1466788	1	110076108	*ALX3*	A/G	0.41	−0.007 (0.001)	3.05 × 10^−16^	0.26	0.01	0.07	0.8629
	So-Youn Shin [7]	rs9842133	3	179946314	*PEX5L*	T/C	0.66	0.006 (0.001)	4.20 × 10^−12^	0.54	0.06	0.06	0.2984
	Karsten Suhre [8]	rs7094971	10	59689806	*SLC16A9*	G/A	0.15	−0.049	3.4 × 10^−14^	0.11	−0.07	0.10	0.4627
	Eugene P Rhee [17]	rs1171617	10	59707424		G/T	0.23	−0.42 (0.04)	5.9 × 10^−26^	0.24	−0.23	0.07	**0.0014**
	Idil Yet [22]						NA	NA	2.3 × 10^−13^				
Citrulline	So-Youn Shin [7]	rs56322409	10	95636205	*ALDH18A1*	T/C	0.63	−0.011 (0.002)	7.81 × 10^−11^	0.92	0.04	0.12	0.7471
											0.02	0.12	0.8796
Creatine	Eugene P Rhee [17]	rs7422339	2	210675783	*CPS1*	A/C	0.31	0.24 (0.04)	2.5 × 10^−11^	0.43	0.19	0.07	**0.0045**
	Bing Yu [9]	rs2433610	15	45393893	15kb from *GATM*	T/C	0.49	NA	9× 10^−12^	0.51	0.01	0.06	0.8755
Glutamine	Karsten Suhre [8]	rs2657879	12	56471554	*GLS2*	G/A	0.19	−0.035	3.1 × 10^−17^	0.06	−0.16	0.13	0.2482
											−0.07	0.14	0.5944
Histidine	Johannes Kettunen [21]	rs7954638	12	95921017	*HAL*	A/C	0.48	−0.08 (0.01)	7.3 × 10^−15^	0.67	−0.09	0.07	0.1863
											−0.06	0.07	0.3814
Inosine	Karsten Suhre [8]	rs494562	6	85407411	*NT5E*	G/A	0.11	0.302	7.4 × 10^−13^	0.39	0.10	0.06	0.0939
Phenylalanine	Michael Inouye [15]	rs1912826	4	186228386	*KLKB1*	G/A	MAF 0.43	NA	3.72 × 10^−12^	0.32	−0.01	0.07	0.8876
Proline	Eugene P Rhee [17]	rs2078743	22	18979346	*PRODH*	A/G	0.09	0.49 (0.06)	2.2 × 10^−14^	0.14	0.12	0.09	0.1968
	Karsten Suhre [8]	rs2023634	22	18984937		G/A	0.09	0.113	2.0 × 10^−22^	0.11	−0.03	0.10	0.7947
	Ayşe Demirkan [19]	rs3213491	22	19177322	*SLC25A1*	A/C	0.95	0.38 (0.11)	7.48 × 10^−4^	0.70	0.10	0.07	0.1402
Serine	So-Youn Shin [7]	rs1163251	1	119667132	*PHGDH*	T/C	0.60	0.019 (0.002)	7.05 × 10^−27^	0.89	0.06	0.10	0.5274
											−0.10	0.10	0.3229
	Karsten Suhre [8]	rs477992	1	119714953		A/G	0.31	−0.051	2.6 × 10^−14^	0.38	−0.02	0.07	0.7487
											−0.12	0.07	0.0632
	So-Youn Shin [7]	rs4947534	7	56011401	*PSPH*	T/C	0.25	−0.018 (0.002)	1.96 × 10^−14^	0.37	−0.04	0.07	0.5866
											−0.10	0.07	0.1450
Tryptophan	So-Youn Shin [7]	rs13122250	4	155887136	*TDO2*	T/C	0.55	0.006 (0.001)	8.95 × 10^−12^	0.13	−0.07	0.10	0.4590
Tyrosine	Tanya M Teslovich [25]	rs28601761	8	125487789	49 kb downstream of *TRIB1*	G/C	0.42	−0.09 (0.02)	8.8 × 10^−9^	0.42	0.04	0.07	0.5151
Urate	Karsten Suhre [8]	rs4481233	4	9954455	*SLC2A9*	T/C	0.19	−0.074	5.5 × 10^−34^	0.08	−0.13	0.11	0.2622
											−0.16	0.12	0.1922

Bold font indicates statistical significance. * Minor allele frequency (MAF) listed if effect allele unknown. ** Standard error (SE) not listed if unavailable. Note: Chr.: chromosome; Pos.: base pair position in human genome built GRCh38; Freq.: frequency; Beta: beta coefficient from linear regression models; SE: standard error from linear regression models; *p*: *p*-value.

**Table 4 metabolites-12-00624-t004:** Genetic associations with metabolites identified in GWAS (*p* < 5 × 10^−8^).

Metabolite	rsID	Chr.	Pos. (GRCh38)	Gene	Effect/ Non-Effect Allele	Effect Allele Freq.	Beta	SE	*p*
1-aminocyclopropane-1-carboxylate	rs112118947	9	114067084	*AMBP*	T/G	0.13	−0.49	0.09	**2.83 × 10^−8^**
1-methylnicotinamide	rs7844962	8	110190121	Intergenic	G/A	0.09	−0.62	0.11	**4.87 × 10^−8^**
3-methyl-2-oxindole	rs6874865	5	152559517	Intergenic	G/A	0.17	−0.48	0.08	**8.66 × 10^−^** ** ^9^ **
Bilirubin	rs9884125	4	183605287	Intergenic	G/A	0.33	0.35	0.06	**3.59 × 10^−8^**
Biliverdin	rs1976391 *	2	233757337	*UGT1A3-10*	G/A	0.40	0.39	0.07	**1.04 × 10^−8^**
Caprylic acid	rs10840643	12	122040948	*BCL7A*	T/C	0.48	0.36	0.07	**4.28 × 10^−8^**
Creatine	rs115281368	5	133290340	*FSTL4*	T/C	0.05	0.80	0.14	**2.62 × 10^−8^**
Creatinine	rs1810668	13	113344465	*GRTP1*	A/G	0.31	0.38	0.07	**1.83 × 10^−8^**
D-gulonic acid gama-lactone	rs2328985 **	13	76682571	*LOC105370266, LOC112268120*	A/C	0.20	−0.40	0.07	**2.58 × 10^−8^**
Glycerate	rs17136208 ***	16	3095047	*ZSCAN10*	C/T	0.05	0.76	0.14	**3.70 × 10^−8^**
Hypotaurine	rs115656245	11	124577645	Intergenic	C/T	0.06	0.72	0.13	**3.98 × 10^−8^**
Hypoxanthine	rs1401798 ****	2	150817357	Intergenic	G/T	0.53	0.37	0.06	**8.80 × 10^−10^**
L-arabitol	rs12603355 *****	17	7829719	*DNAH2*	T/C	0.29	−0.38	0.07	**2.71 × 10^−8^**
Melanin	N/A	1	200189144	*NR5A2*	G/A	0.20	−0.44	0.08	**3.50 × 10^−8^**
N-acetyl-d-tryptophan	rs75313733 ******	6	66785398	Intergenic	C/CT	0.41	−0.38	0.06	**4.70 × 10^−9^**
Palmitoleic acid	rs146744192	19	6299380	Intergenic	T/C	0.19	0.43	0.08	**3.60 × 10^−8^**
Pyridoxamine	rs10170273	2	151664582	*NEB*	C/T	0.34	−0.36	0.06	**3.86 × 10^−8^**
Pyruvate	rs480446	18	60159460	Intergenic	A/G	0.09	0.61	0.11	**3.12 × 10^−8^**
Rac-glycerol 1-myristate	rs11598219	10	68140483	*MYPN*	A/G	0.33	0.38	0.07	**1.56 × 10^−8^**
Sorbate	rs6785673	3	68413982	*TAFA1*	A/C	0.25	−0.40	0.07	**4.57 × 10^−8^**
Trans-cinnamaldehyde	rs10876317	12	52655137	Intergenic	C/T	0.27	0.38	0.07	**4.05 × 10^−8^**
Xanthine	rs4082670	10	11285890	*CELF2*	T/C	0.41	0.34	0.06	**3.80 × 10^−8^**

Bold font indicates statistical significance. * Four other SNPs in Linkage Disequilibrium (LD): rs887829, chr2:233755940, rs4148325, rs4663971. ** One other SNPs in LD: rs9600758. *** One other SNPs in LD: rs116308609. **** Twelve other SNPs in LD: rs7576640, rs6432992, rs10930292, rs7576195, rs6432994, rs1356741, rs11683080, rs2880094, rs2190374, rs1914989, rs6715480, rs1829481. ***** One other SNPs in LD: rs10852892. ****** One other SNPs in LD: rs10806545. Note: Chr.: chromosome; Pos.: base pair position in human genome built GRCh38; Freq.: frequency; Beta: beta coefficient from linear regression models; SE: standard error from linear regression models; *p*: *p*-value.

## Data Availability

Not applicable.

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
