# Peer review of "Crosstalk between Host Genome and Metabolome among People with HIV in South Africa"

_metabolites, 2022, doi:10.3390/metabo12070624_

Round 1

Reviewer 1 Report

Liu et al conducted a GWAS of metabolomics among people with HIV in Africa. The methods and results of GWAS are not described in detail, with much missing information. I cannot determine the soundness of the study without such information. For example, the basic characteristics of the study participants are not presented. It is not clear if some basic covariates like sex and age are adjusted for in the linear regression analysis? The standard QQ- and Manhattan plots are missing. Downstream analysis (e.g. FUMA) to predict how the genome-wide significant SNPs might alter metabolite levels among HIV have not been performed.

Specific comments for each section are included below-

Abstract

“GWAS of 154 plasma metabolites were performed to identify novel genetic associations”. Many genuine associations might have been missed if “novel” association were investigated for just these 154 metabolites?

Introduction

Lines 60-62: Is metabolomics still an emerging field?

Line 78: First appearance of PWH in main text. What does it stand for?

Results

Section 2.1. A table listing the characteristics of the study participants should be included. Basic characteristics include sex, age, medical history (such as history of chronic diseases), history of drug prescription, etc. Any characteristics that might influence the metabolite levels should be included.

Lines 93-96: The authors identified 29 candidate SNP-metabolite associations from published studies. Does it mean that even SNP-metabolite association reported by only one study was included? From Table 1, it is noted that many studies were conducted among Europeans. Is it meaningful to replicate associations of different ethnicities?

Lines 98-100: 24 out of the 36 SNP-metabolite associations were “consistent in direction               regardless of statistical significance”. Is it meaningful if statistical significance is not reached? What threshold did the authors use for the statistical significance of replication? The authors should report the number of SNP-metabolite association with both consistent direction and statistical significance as well.

2.3 GWAS: Some standard quality control metrics should be presented, such as QQ plot and the lambda. Manhattan plot should also be provided.

Downstream analysis of GWAS should be performed to evaluate how the SNPs/genes might affect the metabolite levels in PWH. Otherwise, what are the implications for conducting the GWAS?  

Discussion

The Discussion is too flimsy. Although study participants were PWH, the authors only discussed the UGT1A1 gene, which seems to be the only one related to HIV. How other study findings can be related to HIV? What are the clinical implications of this study? Any strengths and limitations?

Materials and Methods

Lines 203-206: “people with HIV (PWH) who were at least 18 years of age and qualified for anti-retroviral therapy, were enrolled into the HIV AIDS Drug Resistance Surveillance Study (ADReSS). This study was based on a sub-cohort from the ADReSS participants recruited from KwaZulu-Natal, South Africa.” Did the study participants receive anti-retroviral therapy? The therapy might alter the metabolome.

Lines 225-226: Were fasting plasma samples collected? Food intake might alter the metabolome.

Lines 246-248: Only 154 metabolites were included in this “high-resolution” metabolomic study. In recent years, it is common to see over 500 metabolites were measured in GWAS of metabolomics. Based on Table 1, the authors also list a number of metabolic studies that examine >500 metabolites. A recent example is Yin X et al, 2022 (Nat Com), which included over 1000 metabolites.  Can the authors explain why there is such a big difference in the number of metabolites?

GWAS: Are there any covariates adjusted for in the linear regression analysis in addition to the principal components? How about sex and age?

Reviewer 2 Report

I read this article with great interest. The authors conducted a GWAS of plasma metabolites in South African PWH. Not only previously reported genetic associations were replicated, but also some novel genetic associations were identified. Although this work is rather simple, its results are of great value to the researchers in related fields. The paper is well organized and written. So, I support its publication in the journal.  

Reviewer 3 Report

Liu et al present an original artcile entitled "Crosstalk between host genome and metabolome among people with HIV in South Africa". They studied significant gene-metabolite associations in a cohort  of African people of African descent living in Africa suffering from HIV. GWAS of plasma metabolites were performed to identify novel genetic associations. Among the 29 gene-metabolite associations identified in the literature, they found ten in South Africans HIV that were alreay  described in other populations worldwide. They also  identified novel genetic associations using this. GWAS- metabolomics approach.

The article is intesresting and novel. The introduction is too short coenrning the topic. More information should be given to enlightened the reader on the relevency of the study, particularly details on already performed studies.

Why did the authors study 63.7% male subjects?  More details (Table?) shoudl be given about the patients.

 A recapitulative figure would be welcome.
